# Phytase Supplementation of Four Non-Conventional Ingredients Instead of Corn Enhances Phosphorus Utilization in Yellow-Feathered Broilers

**DOI:** 10.3390/ani12162096

**Published:** 2022-08-17

**Authors:** Chengkun Fang, Qifang Yu, Jianhua He, Rejun Fang, Shusong Wu

**Affiliations:** 1College of Animal Science and Technology, Hunan Agriculture University, Changsha 410128, China; 2College of Life Sciences, Hunan Normal University, Changsha 410081, China

**Keywords:** amino acids, broken rice, digestibility, distillers dried grains with soluble, phytase, wheat, wheat bran, yellow broilers

## Abstract

**Simple Summary:**

The serious shortage of feedstuff resources is a prominent problem in the world today, and under the influence of the global food crisis, the price of ingredients has also increased. Due to the usable nutritional value and reasonable price of non-conventional ingredients, making full use of non-conventional ingredients is an important way to alleviate the shortage of feed resources, reduce the cost of livestock and poultry feeding, and improve economic efficiency. However, due to the presence of anti-nutritional factors in unconventional feed ingredients, existing studies suggest that supplementation with feed enzymes, including phytase and xylanase, can enhance feed utilization by eliminating anti-nutritional factors. By evaluating the nutritional composition and use-value of certain non-conventional feed ingredients, an application database can be established that can provide a reference for the extensive use of these resources.

**Abstract:**

The present study was conducted to evaluate the effects of unconventional feedstuff such as wheat, broken rice, distillers dried grains with soluble (DDGS), and wheat bran, replacing 15% of the corn in the basal diet and the supplementation of bacterial phytase on nutrition digestibility. A total of 500 yellow-feathered broilers with similar body weights of 1.65 ± 0.15 kg were divided into 10 dietary treatments with 5 replicates per treatment (5 male and 5 females per cage). The AME and AIDE were significantly higher when supplied with phytase (*p* < 0.01) in the DDGS group. The ileal and total tract digestibility of calcium and phosphorus were significantly increased in the phytase-supplied group (*p* < 0.001). Additionally, the ileal digestibility of CP was increased when phytase was supplemented (*p* < 0.001). The results infer that the wheat, broken rice, DDGS, and wheat bran had no negative effect when replacing 15% corn. Supplementing 0.02% phytase in their diets can effectively optimize nutrient digestibility in yellow broilers.

## 1. Introduction

The expenditure of feeding makes up 65% of the total cost of poultry production, indicating that the exploration of low-cost ingredients is imperative. An increasing amount of animal products, such as meat, eggs, and milk, have been required to meet human consumption due to the rapidly growing population of the world. To address this, large amounts of feed were applied in the animal industry, especially broilers, which subsequently led to a feed deficiency and increase of ingredient cost. Therefore, it is necessary to improve the efficiency of nutrient utilization in broiler chickens.

Given that unconventional feedstuffs have good nutritional value and are reasonably priced, the utilization of unconventional feedstuffs, such as the byproducts obtained from wheat, rice, and other cereal, have received considerable attention [1,2]. However, understanding the effects of the inclusion of wheat, broken rice, DDGS, and wheat bran to broilers’ diets are still unclear. There is no doubt that phosphorus in wheat, broken rice, DDGS, and wheat bran were stored primarily in the form of phytic acid [3], which has been regarded as an anti-nutrient for decades [4]. Therefore, existing studies suggest that supplementation with feed enzymes, including phytase and xylanase, can enhance the feeds’ utilization by eliminating anti-nutritional factors, and also reducing environmental pollution [5]. Babatunde [6] recommends supplementing with phytase at 2000 FYT/kg for 2 days during days 14–21 after broiler chicks hatch. It is reported that phytase supplementation may be related to intermediary phytate degradation products [7]. Gautier et al. [8] also found that phytase influenced growth performance and apparent nutrient digestibility in broiler chicks. Phytase supplementation significantly improved growth, European production index, and economic efficiency, regardless of OC level [9]. Moreover, bacterial phytase at 500 U seems to be more effective than fungal phytase at the same concentration in broilers [10]. Thus, the objective of this study was to evaluate the effects of supplementing bacterial phytase into unconventional feedstuff-replaced diets on the amino acids and energy, crude protein, and calcium and phosphorus digestibility in yellow-feathered broiler chickens.

## 2. Materials and Methods 

This experiment was conducted on the Animal Farming Base of the College of Animal Science and Technology, Hunan Agricultural University (Changsha, China). The experimental protocol (No.2013-06) describing animal usage and care was reviewed and approved by the Committee of Laboratory Animal Management, Animal Welfare of Hunan Agricultural University (Changsha, China) and the Ethical Committee of Hunan Agricultural University. 

### 2.1. Chicken and Experimental Design 

A total of 500 (250 male and 250 female), 80-day old, yellow-feathered broilers with weights of 1.65 ± 0.15 kg (mean ± standard error of the mean) were provided from Ningxian, Wenshi Poultry Company. The broilers were housed in pairs in 39 × 35 × 38 cm wire cages equipped with five ladders with water and feed ad libitum. Every cage was considered an experimental replicate, and each dietary treatment was fed to five replicates (5 male and 5 females per cage). The experiment was performed for 20 days, including 16 days of pre-feeding and 4 days of formal experiment. The pre-feeding period was performed to allow the boilers time to adapt to the environment and diets. During the experimental period, the broilers were randomly assigned to one of either ten treatments based on the requirements of the Nutrition Requirements of Poultry [11] and nutrient requirements of yellow chickens [12]: basal diet consisting of corn–soybean meal (CON); 15% corn-replaced of wheat, broken rice, DDGS, and wheat bran; or 0.02% phytase supplemented with CON, wheat, broken rice, DDGS, and wheat bran. All unconventional feedstuffs were obtained from Tang Ren Shen Group. *E. coli* phytase with 5000 IU/g of vitality was supplied by Guangdong VTR Bio-Tech Co., Ltd. The feed ingredients and their chemical compositions were presented in Table 1 and Table 2, respectively. The ingredient composition and nutritive value of the experimental diets are shown in Table 3. Chickens were fed a powder diet twice daily (9:00 and 16:00).

### 2.2. Sample Collections and Measurements

During the four consecutive days (from the 17th day to the 20th day) of the experimental period, the total tract fecal samples were collected repeatedly every time to measure the apparent metabolic energy (AME), then debris such as dander and feed on the plastic sheets were removed, and the samples were treated with 10% sulfuric acid to fix nitrogen. After the test, the feces samples of each replicate were mixed evenly and stored at −20 °C.

On day 20, 4 broilers (2 males and 2 females) with similar body weight were selected from each cage for the slaughter experiment, and the ileal digesta samples were collected and stored in an aluminum box in the refrigerator at −20 °C. All the digesta samples were freeze-dried and crushed through a 40-mesh sieve to prepare the analytical samples for use. 

### 2.3. Determination Indexes and Method

The sample of feed, ileal digesta, and excreta contents were measured following the Association of Official Analytical Chemists standard procedures [13], with the specific methods as follows: the gross energy (GE) was determined by using an oxygen bomb calorimeter (AN 2001, Accexp Cp., Changsha, China) with benzoic acid as a calibration standard. The nitrogen content was measured by an Automatic Kjeldahl nitrogen analyzer (K1160, Hanon, Jinan, China) and was used to calculate the crude protein content, which was then multiplied by 6.25. 

The acid insoluble ash was measured by following the method of Scott et al. [14].

### 2.4. Calculation of Nutrient Digestibility 

The nutrient digestibility of diets was determined by an endogenous indicator method and calculated according to the following formulae: AME=GEdiet×[100−(AIAdiet/AIAexcreta)×(GEexcreta/GEdiet)×100]
AIDE= GEdiet×[100−(AIAdiet/AIAileal)×(GEexcreta/GEileal)×100]
 AID=100−[AIA diet/AIA excreta×(Nutrition Ileal digesta×Nutrition diet)×100]
ATTD%=100−[AIA diet/AIA excreta×Nutrition excreta×Nutrition diet×100]
where AID% is the percentage of apparent ileal digestibility, ATTD% is the percentage of apparent total tract digestibility, AME is the apparent metabolizable energy, and AIA is the acid-insoluble ash. 

### 2.5. Statisticanalysis

The data were analyzed by the MIXED model of SAS (v9.2, SAS Institute, Cary, NC, USA) to evaluate the treatment effects and compare the test variables that consist of the wheat, broken rice, distillers dried grains with soluble (DDGS), and wheat bran diets that replaced the corn source and the diets supplemented with phytase in 5 × 2 factorial arrangement. The main effects of the wheat, broken rice, distillers dried grains with soluble (DDGS), and wheat bran diets, as well as the diets supplemented with phytase, and their interactions, were tested on each of the determined parameters. A significance effect was declared at *p* < 0.05 and considered a trend at *p* < 0.10. Upon a significant global effect, the treatment means were separated using Tukey’s method.

## 3. Results

### 3.1. AME and AIDE in Different Broiler Diets with or without Added Phytase

The AME and AIDE were increased by the replacement corn source and were phytase-supplemented (*p* < 0.001; Table 4).

### 3.2. Effect of Phytase on Apparent Ileal Digestibility of CP, Calcium and Phosphorus, and Amino Acid in Different Broiler Diets

The ileal digestibility of CP was greater when supplemented with phytase among the corn, wheat, broken rice, DDGS, and wheat bran diets (*p* < 0.001; Table 5), while it was not influenced by the source of ingredient replacement with corn (*p* > 0.05). The digestibility of calcium and phosphorus were higher when supplemented with phytase (*p* < 0.05).

The digestibility of amino acids in the ileal tract was not affected by phytase supplementation. An interaction (*p* < 0.05; Table 6) between replacement corn sources and supplemented phytase was observed for the AID of His, Ile, Leu, Lys, Phe, Thr, Val, Ala, Asp, Gly, Glu, and Ser. The AID of His and Ile on CON was higher than the replacement source (*p* < 0.05), while the other amino acids were lower than wheat, broken rice, DDGS, or wheat bran.

### 3.3. Effect of Phytase on Total Tract Digestibility of CP, Calcium and Phosphorus, and Amino Acids in Different Broiler Diets

As shown in Table 7, the total tract digestibility of calcium and phosphorus was improved when supplemented with phytase (*p* < 0.001), while supplementation did not influence CP (*p* > 0.05). The total tract digestibility of CP and calcium and phosphorus was affected by the interaction between the corn source and phytase supplementation (*p* < 0.001).

An interaction (*p* < 0.05; Table 8) between the replacement corn source and phytase supplementation was observed for the ATTD of Arg, His, Ile, Leu, Lys, Phe, Thr, Val, Ala, Asp, Gly, Glu, Ser, and Tyr, while it was not observed for Met (*p* > 0.05). The ATTP of the amino acids were influenced by the replacement corn source (*p* < 0.05).

## 4. Discussion

### 4.1. AME and AIDE in Different Broiler Diets with or without Added Phytase

The purpose of this study was to investigate the impact of supplemented phytase and the replacement of corn with different ingredients on the nutrient digestibility in yellow-feathered broilers. The chemical composition of the ingredients (wheat, broken rice, DDGS, wheat bran) were measured before the experiment and the result values were similar with previous studies [15,16]. A lack of metabolizable energy and calcium and phosphorus will affect the growth and development of animals. The AME and AIDE of the ingredients were greater when supplemented with phytase according to numerous previous studies [17,18,19]. In the present study, the AME and AIDE were improved when supplemented with phytase. Moreover, a basal diet replaced by wheat, DDGS, and wheat bran were shown to increase the AME and AIDE. Ravindran et al. [20] found that the addition of phytase and xylanase alone in a wheat diet could improve the AME by 9.7% and 5.3%, respectively, while the addition of both enzymes could increase the AME by 19%. Gallardo [21] found that the substitution of corn with wheat bran and the addition of 500 FTU/kg of phytase improved the AME of broilers from 5.84 MJ/kg to 6.16 MJ/kg. 

The AME of broilers in the wheat, DDGS, and wheat bran group were higher than the soybean meal feeding group, while the value of the broken rice feeding group was lower. However, the AIDE of DDGS and wheat bran were higher than that of soybean meal, while the AIDE in wheat and broken rice were lower than that of soybean meal. Ban et al. [16] showed that the AME of DDGS was greater when supplemented with 40% corn, which is consistent with our results; the reason may be related to a higher ME in DDGS than that in corn. 

### 4.2. Effect of Phytase on Apparent Ileal Digestibility of CP, Calcium and Phosphorus, and Amino Acid in Different Broiler Diets

Phytate phosphorus was largely unavailable in broilers, although they take up 60–80% of the total phosphorus in most grains [22]. Supplementation with phytase was considered to be an effective strategy to improve the utilization of phosphorus in grains [23]. Previous studies have reported that the ileal digestibility of CP, calcium and phosphorus, and amino acids were improved when supplemented with phytase in broiler chickens [24,25,26] and pigs [27]. Phytase supplementation tended to increase the standardized total tract digestibility of P in high-protein sunflower meal fed to pigs by 100% [28]. In the current study, we found that supplemented phytase in different diets would improve the ileal digestibility of calcium from 0.86% to 35.24% and improve the ileal digestibility of phosphorus from 14.95% to 30.73%. These results are in agreement with previous studies [24]. However, no significant effects were observed on the ileal digestibility of amino acids when additional phytase was applied. There is a dispute about the effect of phytase on the ileal digestibility of amino acids. Amerah and Ravindran confirmed that supplementation with 1000 FTU/kg of phytase improved the AID of amino acids [20,25]; whereas Kong et al. [29] reported that phytase supplementation had no effect on the ideal digestibility of amino acids in canola meal. It is also reported that phytase effects on AA digestibility were lower but significant [30]. Therefore, the effect of phytase on amino acid digestibility is closely related to the diet.

The digestibility of CP in the ileal tract was not influenced by the 15%-replaced ingredient sources in our results, while the digestibility of amino acids was significant. In the present study, the level of CP is 16.55%. The reasons for some differences are unclear, but they may be related to the amino acid composition of ingredients and intact phytase of different feedstuffs. 

### 4.3. Effect of Phytase on Total Tract Utilization of CP, Calcium and Phosphorus, and Amino Acids in Different Broiler Diets

Phytase can improve the digestibility of protein and phosphorus, as reported by Ravindran et al. [20] and Beeson et al. [31]. Moreover, supplemented phytase can hydrolyze phytic acid; release divalent cations, proteins, and amino acids chelated with phytic acid; eliminate the anti-nutritional properties of phytic acid; and thus increase the digestibility of these nutrient substances in monogastric animals [32]. The utilization of calcium improves when supplemented with phytase [31]. However, the ratio of calcium to phosphorus or available phosphorus also affected the utilization of calcium. Generally, phytase works better at lower calcium levels and with high phytic acid diets [5]. In this study, the utilization of calcium, phosphorus, and CP were increased when supplemented with phytase, which was consistent with the previous study [33,34]. In fact, in the current study, the proportion of calcium and phosphorus in the raw materials of the diet were not considered, and the small amounts of calcium and phosphorus that phytase itself contains was neglected as well, which is a deficiency of this experiment.

The CP and amino acids were not influenced by phytase supplementation in our results, which are not in agreement with Kong et al. [29]. We observed that amino acids differ among different ingredients. These responses refer to the crude protein and amino acid composition, among others.

## 5. Conclusions

In conclusion, our study suggested that supplementation with 200 mg/kg of *E. coli* phytase with 5000 IU/g can improve the energy, crude protein, and calcium and phosphorus digestibility when used with a wheat, broken rice, DDGS, or wheat bran diet that replaces 15% of the total corn of the basal diet. However, the digestibility of amino acids was not significantly influenced after feeding with phytase. In addition, the current study provided information about the utilization of unconventional feedstuffs on yellow-feathered broilers.

## Figures and Tables

**Table 1 animals-12-02096-t001:** The characteristics of the feedstuff samples.

Name	Place of Origin	Characteristics Description
Wheat	Henan	Rough grains, mature, mixed winter wheat
Broken rice	Hunan	Early indica rice with shell removed, pale yellow with green, mature, Chanyou 63
DDGS distillers dried grains with soluble	North China	Domestic low oil corn DDGS, light yellow
Wheat bran	Henan	Gray rough, finely ground, with a small amount of wheat husks

**Table 2 animals-12-02096-t002:** Chemical composition (value analyzed) of the experimental ingredients evaluated in the experiments.

Items	Wheat	Broken Rice	DDGS ^1^	Wheat Bran
Dry matter, %	85.98	86.58	87.92	87.51
Gross energy, Kcal/g	3.22	3.53	2.27	1.49
Crude protein, %	12.74	8.79	26.36	16.09
Crude fiber, %	1.90	0.70	6.60	6.50
Calcium, %	0.13	0.03	0.05	0.11
Phosphorus, %	0.51	0.27	0.48	0.70
Essential amino acids ^2^				
Arg, %	0.55	0.48	1.12	1.19
His, %	0.44	0.31	1.07	0.64
Ile, %	0.43	0.31	0.95	0.51
Leu, %	0.84	0.65	3.05	1.07
Lys, %	0.33	0.29	0.79	0.67
Met, %	0.16	0.13	0.28	0.12
Phe, %	0.53	0.39	1.25	0.61
Thr, %	0.37	0.29	1.05	0.56
Val, %	0.54	0.46	1.25	0.76
Non-essential amino acids ^3^				
Ala, %	0.51	0.45	2.00	0.91
Asp, %	0.65	0.68	1.62	1.16
Glu, %	3.38	1.45	4.17	3.04
Gly, %	0.49	0.35	1.04	0.88
Pro, %	1.21	0.30	2.08	1.00
Tyr, %	0.18	0.15	1.00	0.50
Ser, %	0.59	0.41	1.37	0.76

^1^ DDGS: distillers dried grains with soluble. ^2^ Arg: arginine; His: histidine; Ile: isoleucine; Leu: leucine; Lys: lysine; Met: methionine; Phe: phenylalanine; Thr: threonine; Val: valine. ^3^ Ala: alanine; Asp: aspartic; Glu: glutamic; Gly: glycine; Pro: proline; Tyr: tyrosine; Ser: serine.

**Table 3 animals-12-02096-t003:** Ingredient composition and nutritive value of the experimental diets.

Items	Basal Diet		Wheat		Broken Rice		DDGS ^1^		Wheat Bran	
	P^− 2^	P^+ 3^	P^−^	P^+^	P^−^	P^+^	P^−^	P^+^	P^−^	P^+^
Corn, %	66.04	66.02	56.40	56.40	56.00	56.00	59.04	59.02	56.51	56.49
Soybean meal (43%), %	23.80	23.8	23.54	23.52	22.28	22.26	18.34	18.34	22.00	22.00
Soybean oil, %	4.00	4.00	3.70	3.70	6.20	6.20	5.21	5.21	3.80	3.80
Stone powder, %	1.13	1.13	1.13	1.13	1.13	1.13	1.13	1.13	1.13	1.13
Lysine, %	0.10	0.10	0.09	0.09	0.10	0.10	0.19	0.19	0.13	0.13
Methionine, %	0.10	0.10	0.10	0.10	0.12	0.12	0.08	0.08	0.10	0.10
Zeolite powder, %	1.50	1.50	1.69	1.69	0.20	0.20	1.50	1.50	1.50	1.50
Wheat, %	-	-	10.02	10.02	-	-	-	-	-	-
Broken rice, %	-	-	-	-	10.64	10.64	-	-	-	-
DDGS, %	-	-	-	-	-	-	11.18	11.18	-	-
Wheat bran, %	-	-	-	-	-	-	-	-	11.5	11.5
Phytase, %	-	0.02	-	0.02	-	0.02	-	0.02	-	0.02
Calcium hydrogen phosphate, %	1.33	1.33	1.33	1.33	1.33	1.33	1.33	1.33	1.33	1.33
Premix ^4^	2.00	2.00	2.00	2.00	2.00	2.00	2.00	2.00	2.00	2.00
Total, %	100	100	100	100	100	100	100	100	100	100
Nutrient level ^5^										
Metabolic energy, kcal/kg	3.06	3.06	3.06	3.06	3.06	3.06	3.05	3.05	3.04	3.04
Crude protein, %	16.55	16.55	16.55	16.55	16.55	16.55	16.55	16.55	16.54	16.54
Calcium, %	0.81	0.81	0.81	0.81	0.81	0.81	0.81	0.81	0.82	0.82
Total phosphorus, %	0.57	0.57	0.58	0.58	0.63	0.63	0.60	0.60	0.58	0.58
Effective phosphorus, %	0.38	0.38	0.38	0.38	0.39	0.39	0.40	0.40	0.38	0.38
Lysine, %	0.88	0.88	0.88	0.88	0.88	0.88	0.88	0.88	0.88	0.88
Methionine, %	0.35	0.35	0.35	0.35	0.35	0.35	0.35	0.35	0.35	0.35
Threonine, %	0.65	0.65	0.64	0.64	0.63	0.63	0.63	0.63	0.62	0.62
Tryptophan, %	0.19	0.19	0.20	0.20	0.20	0.20	0.17	0.17	0.19	0.19

^1^ DDGS: distillers dried grains with soluble. ^2^ P^−^: no supplementation with phytase; ^3^ P^+^: supplementation with phytase. ^4^ Premix provided the following per kilogram of diet: vitamin A 10,000 IU; vitamin D3 2750 IU; vitamin E 20 IU; vitamin K3 2 mg; vitamin B1 1.5 mg; riboflavin 6 mg; pantothenic acid 12 mg; niacin 20 mg; vitamin B6 2.5 mg; vitamin B12 12 ug; choline 500 mg; Mn 75 mg; Zn 75 mg; Fe 95 mg; Cu 10 mg; I 0.6 mg; Se 0.3 mg. ^5^ Calculated according to NRC (1994).

**Table 4 animals-12-02096-t004:** AME and IDE in different broiler diets with or without added phytase.

Items		Basal Diet	Wheat	Broken Rice	DDGS ^1^	Wheat Bran	SEM ^6^	*p*-Values ^7^		
								S	P	P * S
AME ^2^, kcal/kg	P^− 4^	3227	3399	3036	3451	3391	52.5	<0.001	<0.001	<0.001
	P^+ 5^	3394	3485	3104	3544	3401				
AIDE ^3^, kcal/kg	P^−^	2967	2899	2926	3379	3218	47.8	<0.001	<0.001	<0.001
	P^+^	3078	2912	2981	3416	3369				

^1^ DDGS: distillers dried grains with soluble. ^2^ AME: apparent metabolizable energy; ^3^ AIDE: apparent ileal digestible energy. ^4^ P^−^: no supplementation with phytase; ^5^ P^+^: supplemented with phytase. ^6^ SEM: standard error of the mean. ^7^ S: the source of ingredient replacement with 15% corn; P: phytase; P * S: interaction between source of ingredient replacement with 15% corn and supplemented with phytase.

**Table 5 animals-12-02096-t005:** Effect of diet supplemented with phytase on apparent ileal digestibility of CP and calcium and phosphorus in broilers.

Items		Basal Diet	Wheat	Broken Rice	DDGS ^2^	Wheat Bran	SEM ^3^	*p*-Values ^4^		
								S	P	P * S
CP ^1^, %	P^− 5^	76.2	75.8	69.5	78.3	81.7	1.602	0.9515	<0.001	<0.001
	P^+ 6^	80.2	79.3	72.4	81.4	83.4				
Calcium, %	P^−^	50.7	42.0	42.7	46.7	51.4	1.79	<0.001	<0.001	<0.001
	P^+^	63.9	56.8	48.6	47.1	52.8				
Phosphorus, %	P^−^	41.0	48.7	36.8	47.2	44.3	4.126	<0.001	<0.001	<0.001
	P^+^	53.6	57.7	42.3	58.3	52.7				

^1^ CP: crude protein; ^2^ DDGS: distillers dried grains with soluble. ^3^ SEM: standard error of the mean. ^4^ S: the source of ingredient replacement with 15% corn; P: phytase; P * S: interaction between source of ingredient replacement with 15% corn and supplemented with phytase. ^5^ P^−^: no supplementation with phytase; ^6^ P^+^: supplemented with phytase.

**Table 6 animals-12-02096-t006:** Effect of diet supplemented with phytase on apparent ileal digestibility of amino acids in broilers.

Items ^1^		Basal Diet	Wheat	Broken Rice	DDGS ^2^	Wheat Bran	SEM ^3^	*p*-Values ^4^		
								S	P	P * S
Arg	P^− 5^	82.1	75.7	80.8	78.1	85.7	2.19	0.025	0.926	0.078
	P^+ 6^	85.1	83.5	84.0	80.8	89.2				
His	P^−^	82.3	78.2	72.7	69.8	75.4	3.33	0.006	0.415	0.008
	P^+^	84.7	85.6	77.9	75.7	82.1				
Ile	P^−^	78.3	82.1	55.8	75.0	79.9	1.62	0.007	0.951	0.035
	P^+^	82.4	87.5	59.8	76.3	85.7				
Leu	P^−^	87.9	83.4	64.2	81.2	81.7	3.03	0.007	0.954	0.027
	P^+^	88.8	88.7	69.5	83.7	86.5				
Lys	P^−^	72.9	71.7	66.3	73.9	82.3	2.20	0.017	0.928	0.050
	P^+^	75.4	79.5	69.7	73.1	87.7				
Met	P^−^	75.3	76.9	71.3	84.4	88.8	1.60	0.091	0.629	0.086
	P^+^	76.4	80.1	73.8	90.2	91.7				
Phe	P^−^	82.5	83.5	55.7	78.1	81.7	1.61	0.006	0.998	0.023
	P^+^	84.2	88.8	60.8	81.0	86.7				
Thr	P^−^	66.6	66.4	52.4	78.6	71.9	2.70	0.018	0.624	0.033
	P^+^	70.4	76.0	56.1	83.1	79.0				
Val	P^−^	76.4	77.6	57.0	71.6	78.9	2.20	0.013	0.872	0.045
	P^+^	79.5	84.4	60.2	86.8	84.3				
Ala	P^−^	85.0	74.7	71.5	87.1	81.3	1.80	0.006	0.758	0.015
	P^+^	86.1	82.0	76.2	91.0	86.3				
Asp	P^−^	72.0	70.8	60.6	77.1	76.3	3.34	0.004	0.951	0.012
	P^+^	75.4	79.3	65.4	83.8	82.7				
Gly	P^−^	67.8	75.0	67.6	69.2	75.4	1.77	0.008	0.942	0.020
	P^+^	71.9	82.7	73.3	71.2	82.4				
Glu	P^−^	88.0	91.3	73.0	88.1	86.0	4.53	0.003	0.821	0.013
	P^+^	88.6	94.4	76.9	92.4	90.2				
Ser	P^−^	73.3	78.0	60.8	84.5	76.1	2.44	0.012	0.673	0.020
	P^+^	77.5	84.9	67.5	87.9	84.1				
Tyr	P^−^	74.3	73.3	54.8	89.9	80.3	1.92	0.031	0.574	0.087
	P^+^	76.3	81.3	60.5	91.6	84.5				

^1^ Arg: arginine; His: histidine; Ile: isoleucine; Leu: leucine; Lys: lysine; Met: methionine; Phe: phenylalanine; Thr: threonine; Val: valine; Ala: alanine; Asp: aspartic; Gly: glycine; Glu: glutamic; Ser: serine; Tyr: tyrosine. ^2^ DDGS: distillers dried grains with soluble. ^3^ SEM: standard error of the mean. ^4^ S: the source of ingredient replacement with 15% corn; P: phytase; P * S: interaction between source of ingredient replacement with 15% corn and supplemented with phytase; SEM: standard error of the mean. ^5^ P^−^: no supplementation with phytase; ^6^ P^+^: supplemented with phytase.

**Table 7 animals-12-02096-t007:** Effect of diet supplementary with phytase on apparent total tract digestibility of CP and calcium and phosphorus in broilers.

Items		Basal Diet	Wheat	Broken Rice	DDGS ^1^	Wheat Bran	SEM ^2^	*p*-Values ^3^		
								S	P	P * S
CP ^4^, %	P^− 5^	72.1	75.8	66.5	76.5	74.4	0.28	<0.001	0.1504	0.007
	P^+ 6^	74.2	77.4	70.9	79.7	77.4				
Calcium, %	P^−^	51.2	57.3	51.2	52.8	58.2	1.936	<0.001	<0.001	<0.001
	P^+^	66.1	59.3	53.1	55.4	63.2				
Phosphorus, %	P^−^	53.6	63.3	52.3	63.5	58.4	4.007	<0.001	<0.001	<0.001
	P^+^	68.3	76.2	64.2	71.2	69.7				

^1^ DDGS: distillers dried grains with soluble. ^2^ SEM: standard error of the mean. ^3^ S: the source of ingredient replacement with 15% corn; P: phytase; P * S: interaction between source of ingredient replacement with 15% corn and supplemented with phytase. ^4^ CP: crude protein. ^5^ P^−^: no supplementation with phytase; ^6^ P^+^: supplemented with phytase.

**Table 8 animals-12-02096-t008:** Effect of diet supplementary with phytase on apparent total tract digestibility of amino acids in broilers.

Items ^1^		Basal Diet	Wheat	Broken Rice	DDGS ^2^	Wheat Bran	SEM ^3^	*p*-Values ^4^		
								S	P	P * S
Arg	P^− 5^	86.2	77.6	70.3	69.0	80.8	0.5690	<0.001	0.690	<0.001
	P^+ 6^	88.3	84.8	81.7	75.4	84.9				
His	P^−^	82.6	68.0	73.8	69.6	82.4	1.2570	0.006	0.784	0.001
	P^+^	84.3	77.3	79.1	74.9	85.4				
Ile	P^−^	78.7	73.4	52.5	71.5	85.1	0.71	<0.001	0.923	<0.001
	P^+^	82.3	79.8	54.8	76.5	89.6				
Leu	P^−^	82.2	75.9	66.7	66.9	74.2	1.05	<0.001	0.647	<0.001
	P^+^	84.1	82.8	68.4	73.4	77.2				
Lys	P^−^	84.2	61.7	68.2	74.9	78.9	0.73	<0.001	0.828	<0.001
	P^+^	85.9	70.4	71.2	80.1	85.4				
Met	P^−^	91.4	80.2	77.3	87.2	77.0	0.13	<0.001	0.754	0.0854
	P^+^	91.3	83.6	83.4	90.5	85.4				
Phe	P^−^	81.0	77.3	57.9	76.6	84.6.	0.61	<0.001	0.711	<0.001
	P^+^	83.3	84.1	63.1	81.4	85.6				
Thr	P^−^	78.0	54.5	52.0	63.8	77.2	1.79	<0.001	0.870	<0.001
	P^+^	79.7	66.7	57.8	74.1	82.4				
Val	P^−^	77.7	70.5	58.7	60.0	79.8	0.78	<0.001	0.864	<0.001
	P^+^	81.2	78.0	61.4	72.3	87.4				
Ala	P^−^	82.2	54.4	71.5	87.1	74.9	0.88	<0.001	0.675	<0.001
	P^+^	83.2	56.2	76.2	91.0	80.7				
Asp	P^−^	78.9	59.8	61.9	77.1	72.6	1.35	<0.001	0.865	<0.001
	P^+^	80.9	65.4	66.3	83.8	79.3				
Gly	P^−^	75.9	61.5	66.6	69.2	78.4	1.16	<0.001	0.760	<0.001
	P^+^	77.9	64.8	71.3	71.2	84.2				
Glu	P^−^	85.7	88.5	72.7	88.1	75.5	1.92	<0.001	0.619	<0.001
	P^+^	87.0	91.7	74.1	92.4	79.4				
Ser	P^−^	79.4	72.4	61.5	84.5	85.3	1.75	<0.001	0.824	<0.001
	P^+^	79.7	79.8	68.3	87.9	86.1				
Tyr	P^−^	80.2	75.4	55.2	89.9	77.3	0.72	0.001	0.407	0.003
	P^+^	82.1	80.8	59.8	91.6	80.1				

^1^ Arg: arginine; His: histidine; Ile: isoleucine; Leu: leucine; Lys: lysine; Met: methionine; Phe: phenylalanine; Thr: threonine; Val: valine; Ala: alanine; Asp: aspartic; Gly: glycine; Glu: glutamic; Ser: serine; Tyr: tyrosine. ^2^ DDGS: distillers dried grains with soluble. ^3^ SEM: standard error of the mean. ^4^ S: the source of ingredient replacement with 15% corn; P: phytase; P * S: interaction between source of ingredient replacement with 15% corn and supplemented with phytase; ^5^ P^−^: no supplementation with phytase; ^6^ P^+^: supplemented with phytase.

## Data Availability

Not applicable.

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
