# Peer review of "Phytase Supplementation of Four Non-Conventional Ingredients Instead of Corn Enhances Phosphorus Utilization in Yellow-Feathered Broilers"

_animals, 2022, doi:10.3390/ani12162096_

Round 1

Reviewer 1 Report

This study investigated the effect of phytase supplementation in four non-conventional ingredients instead of the corn enhances phosphorus utilization in yellow-feathered broilers. Overall, the results are interesting, and the research topic falls within the scope of the Journal. However, the following questions should be addressed before the manuscript can be accepted for publication.

1. The usage of phytase in animal production is common in livestock and poultry production, please highlight the novelty of the present study.

2. The reason why the four unconventional ingredients was selected for experiment should be clearly stated.

3. Section 3.1 is too short to be a section, authors can briefly state the meaning of results, or combined it with other sections.

4. Please add the full name of abbreviation in the notes of tables.

5. Line 94, 4 broilers (2 males and 2 females) with similar body weight were selected, from each replicates, or each group?

6. Line 49 and 52: Reference number should be used after author’s name.

7. Please carefully check the format of all references.

Reviewer 2 Report

This study of Fang et al. investigated the influence the effects of unconventional feedstuff replaced 15% of corn in basal diet and the supplementation of phytase on nutrition digestibility using 500 yellow-feathered broilers with similar body weight of 1.65 ± 0.15 kg, which were distributed to 10 dietary treatments with 5 replicates each.  The result showed that supplementary 0.02% phytase in the diets can effectively optimize the nutrient digestibility in yellow broilers.  The use of phytase in poultry nutrition has been initiated 20 years with many successfully results and practical application to animal diets, thus the novelty of this study must be emphasis  for acceptance  and here is my comments  which could be summarized in the following:

1.      In the introduction section, please emphasis on the added value/novelty of this research and how it could contribute to improve production of yellow feathered broilers.

2.       Here some references could be of added value to your Ms in L 39:

Al-Harthi Mohammed A., Youssef A. Attia, Ali S. Al-Shafey,  and Mohamed F. El-Gendy (2020).  Impact of  phytase on improving the utilisation of pelleted broiler diets containing olive by-products. Ita J Animal Sci.  19: 310-318. DOI: 10.1080/1828051X.2020.1740896

Attia, Y. A., F. Bovera, A. E. Abd El-Hamid, A. E. Tag El-Din, M. A. Al-Harthi and A. S. El-Shafy (2016). Effect of zinc bacitracin and phytase on growth performance, nutrient digestibility, carcass and meat traits of broilers. J. of Anim. Physiology and Anim. Nutrition 100:485-491. DOI: 10.1111/jpn.12397.

3.      L 62, Plz add experimental protocol number.

4.       L 77, Plz indicate the basis of choosing 15% corn-replaced level by unconventional feed resources.

5.       L 77-79, what kind of phytase used E. coil, SN, phytase and what the basis of choosing this level.

6.      What it is the guideline for feeding yellow feather chinses chickens, is there are any references? Plz report in L 81-82

7.      L 82, what form of feed did the chickens offered?

8.      L 93, fecal samples were collected, what type of collection, total gut/tract, ileal digesta collection, plz indicate?

9.      Table 2, Ingredient composition and nutritive value of the experimental diets, why the authors did not take advantage of P and Ca levels (0.1% about for each when phytase was added) authors have to explain this in details, phytase are known to work good at lower calcium level and high phytic acids diets  doi:10.3390/ani9050280.

10.  The statistical model and the experiment unit should be added to the statistical analyses section, L 122

11.  I see the intact phytase of feedstuffs has a major effect on the effect of supplemented phytase reported in this study for example the  AME of wheat bran increased by 10 kcal while in other feedstuffs increased by 70-150 kcal, This due to high intact phytase of wheat bran 1500 U/kg and for this your results and discussion have to be re-evaluated due to different nature and phytase content of different feed stuffs.

12.   L 195-196, please quantities the improvements in the ileal digestibility of calcium, phosphorus

13.   L 206;  and /or intact phytase of different feedstuffs

14.  All abbreviation used in the Tables should be listed under the table, tables must be self-explanation.

15.   L 223,  plz add supplemented dose and type of phytase

16.  The references  must be updated  by 2021 and 2022 published articles  

Reviewer 3 Report

Dear Editor,
Here is my review on the manuscript with the number Animals-1842426
Thanks for the cooperation

Reviewer 4 Report

Dear authors,
The subject of the article is very interesting. However, major corrections need to be made before considering it for publication in Animals.
1- The "simple summary" and the "abstract" are similar. You should look at the instructions in Animals to adapt them.
2- The introduction needs contextualization. The objectives of your article should be explained after introducing your work.
3- Material and Methods: In the Material and Methods section, I have many remarks:
- Concerning the distribution of animals: was it one male and one female per cage; how did you distribute the experimental groups on the different cages? What is the experimental unit? What about your sample size?
- The unit used for live weight is the kilogram. Are your weighings not accurate? The average weight is 1.65 ± 0.15 kg. It would be more interesting to give the mean weight, median, maximum and minimum values by experimental group.
- What are the justifications for the choice of this broiler strain and the age at the beginning of the study (80 days)? Also, why did your study take place over a period of 20 days (16 days pre-feeding and 4 days experimental)?
4-Results: Add below the tables the meanings of the abbreviations (S, P, SEM,...)
5-Concerning the discussion, it would be appropriate to explain the observed results and not just compare them to the literature.
Yours sincerely.

Round 2

Reviewer 2 Report

The authors have responded adequality to the review comments  and I see the Ms Can be published now after some change such use italic for bacterial name such as E. Coli ............. etc.....

Author Response

Dear reviewer,

Thanks for your great work, I have modified my manuscript.

Reviewer 3 Report

Dear editor

All recommendations were done.

REGARDS

Author Response

Dear reviewer,

Thanks for your great work.

Reviewer 4 Report

Dear authors,

thank you for the improvements made to your article.

I wish you all the best for the future.

Best regards,

Author Response

Dear reviewer,

Thanks for your great work.

Best regards!